# Role of the Dorsal Raphe Nucleus in Pain Processing

**DOI:** 10.3390/brainsci14100982

**Published:** 2024-09-28

**Authors:** Huijie Zhang, Lei Li, Xujie Zhang, Guanqi Ru, Weidong Zang

**Affiliations:** 1Department of Anatomy, School of Basic Medical Sciences, Zhengzhou University, Zhengzhou 450001, China; z15225072302@163.com (H.Z.); zdefylilei@126.com (L.L.); 2Department of Traditional Chinese Medicine, Henan University of Chinese Medicine, Zhengzhou 450046, China; z19836406109@163.com; 3Department of Medical Sciences, Zhengzhou University, Zhengzhou 450001, China; ruguanqi@126.com

**Keywords:** dorsal raphe nucleus, neurons, pain, interaction

## Abstract

The dorsal raphe nucleus (DRN) has gained attention owing to its involvement in various physiological functions, such as sleep–awake, feeding, and emotion, with its analgesic role being particularly significant. It is described as the “pain inhibitory nucleus” in the brain. The DRN has diverse projections from hypothalamus, midbrain, and pons. In turn, the DRN is a major source of projections to diverse cortex, limbic forebrain thalamus, and the midbrain and contains highly heterogeneous neuronal subtypes. The activation of DRN neurons in mice prevents the establishment of neuropathic, chronic pain symptoms. Chemogenetic or optogenetic inhibition neurons in the DRN are sufficient to establish pain phenotypes, including long-lasting tactile allodynia, that scale with the extent of stimulation, thereby promoting nociplastic pain. Recent progress has been made in identifying the neural circuits and cellular mechanisms in the DRN that are responsible for sensory modulation. However, there is still a lack of comprehensive review addressing the specific neuron types in the DRN involved in pain modulation. This review summarizes the function of specific cell types within DRN in the pain regulation, and aims to improve understanding of the mechanisms underlying pain regulation in the DRN, ultimately offering insights for further exploration.

## 1. Structure of Dorsal Raphe Nucleus

The dorsal raphe nucleus (DRN), located in the ventromedial part of the midbrain periaqueductal gray, has a fan-shaped structure that is symmetrically distributed along the midline. It is highly heterogeneous in anatomical distribution, molecular markers, synaptic connectivity, and function and consists of various cell types, mainly including serotonergic neurons (5-HT neurons), GABAergic neurons (GABA neurons), dopaminergic neurons (DA neurons), glutamatergic neurons, and peptidergic neurons [1,2]. Among these, the 5-HT neurons are the most abundant, constituting approximately two-thirds of the neurons in the DRN [3,4]. They exhibit a clustered distribution and are predominantly found in the midline region. GABA neurons are second, accounting for about 25%. They predominantly are scattered on both sides of the wing [5]. DRN neurons exhibit heterogeneity and corelease neurotransmitters. For example, DRN 5-HT neurons corelease glutamate and serotonin [6], and some DA neurons co-release dopamine and glutamate [1]. In addition, the presence of glutamic acid decarboxylase 1 (GAD1) and GAD2, which are enzymes responsible for GABA synthesis, in 5-HT and glutamate neurons further demonstrates how the DRN neurons synthesize and release GABA as a neurotransmitter [7] (Figure 1).

## 2. Neurons within DRN Involved in Pain Control

As early as 1993, the DRN was described as the “pain inhibitory nucleus” in the brain [8]. A recent study found that implanting microelectrodes into the DRN and utilizing deep brain stimulation provided analgesia in patients [9,10], emphasizing the pivotal regulatory function of the DRN in pain management.

### 2.1. 5-HT Neurons

5-HT neurons within DRN are the main source of forebrain 5-HT. In previous research, 5-HT neurons have been commonly labeled using markers such as tryptophan hydroxylase (*TPH*), serotonin transporter (*SERT*), *Pet1*, or *Sl6a4* expression genes. 5-HT released by these neurons can regulate the input of pain signals and influence pain processing. During inflammation, it interacts with proinflammatory factors and is introduced along the sensory afferent fibers (Aδ and C fibers), releasing nociceptive neurotransmitters to the dorsal horn of the spinal cord. The pain signals are then transmitted through ascending pathways to the cortex [11]. Studies have shown that the 5-HT knockout mice could increase pain sensitivity and reduced analgesic effects of opioids [12,13]. Additionally, acute traumatic stimulation has been reported to increase the activity of 5-HT neurons in the DRN [14], suggesting that 5-HT neurons in DRN may be an important target for pain treatment. Selective serotonin reuptake inhibitors (SSRIs) are the first-line treatment for depression clinically by blocking serotonin reuptake. Although SSRI have been suggested as an alternative treatment for chronic pain, the safety and effectiveness of SSRIs for pain condition is still uncertain [15].

5-HT receptors comprise 7 families (5-HT1-5-HT7) and approximately 15 receptor subtypes, which modulate neuronal activity [16,17]. Research reported that 5-HT modulates analgesic effects through 5-HT1 [18,19], 5-HT2 [20], and 5-HT7 [21] receptors, which are coupled to the Gq or Gi protein to reduce levels of cAMP and increase IP3 and DG levels. The 5-HT1A receptor appears to be involved in a variety of pain and mood disorders [22]. Hirvonen found an inverse correlation between 5-HT1A receptors binding potential and neuroticism [23]. 5-HT3 receptors depolarize the neuronal membrane [24]. In the central regulation of descending pain facilitation/inhibition from the DRN, various brain regions, such as the cortex, thalamus, and midbrain, are involved in the central regulation of descending pain facilitation/inhibition of DRN [25]. In a sciatic nerve ligation model, the activity of DRN 5-HT neurons increased and the levels of 5-HT released in the PFC increased. Optical stimulation of DRN^5-HT^–PFC alleviated sleep dysregulation induced by chronic pain [26]. Some research found a neural circuit between DRN and ACC modulated chronic pain-induced dysfunction. Activation of the ACC-DRN^5-HT^ pathway reversed chronic pain-induced anxiety-like behaviors [27], and activation of the DRN^5-HT^–ACC pathway reversed sociability deficits [28]. Studies have measured serotonin release in the CeA and ACC. The neural activity of 5-HT neurons increased after formalin injection and increased 5-HT concentrations in the CeA and ACC [29]. In mice with spared nerve injury, the firing of 5-HT neurons in the DRN decreased and resulted in the disinhibition of CeA^SOM^ neurons. Activation of the DRN^5-HT^–CeA^SOM^ pathway relieved pain and depression-like behaviors. The study also used resting state fMRI to assess the neural circuit mechanisms for patients with chronic pain comorbid with depressive symptoms. The results showed that the functional connectivity of the DRN–CM pathway was decreased [30]. One such region is the lateral habenula (LHb), which can enhance the activity of 5-HT neurons in DRN through damaging LHb. This leads to an increased pain threshold in rats with neuropathic pain models [31]. These findings showed that 5-HT neurons in the DRN encode chronic pain by distinct pathway (Figure 2).

5-HT, referred to as the “mood boosters”, contributes to chronic pain comorbid anxiety and depression [32]. In inflammatory pain models and neuropathic pain models, the 5-HT levels in hippocampal and cortical regions increased significantly in the brain and the mice showed anxiety and depression-related behaviors, indicating a connection between reduced 5-HT levels and the development of anxiety and depression in the context of chronic pain [33,34]. Additionally, two brain regions, the bed nucleus of the stria terminalis (BNST) and the ventral tegmental area (VTA), are involved in encoding negative emotions and reward information, respectively [7]. These regions have been implicated in the modulation of anxiety and depression related to chronic pain. Furthermore, many small molecules have been identified that can influence the 5-HT system and participate in the regulation of pain-related negative emotions. For example, prostaglandin E2 [35], adiponectin [36], and cannabidiol [37,38] have been found to regulate the release of serotonin through 5-HT receptors. These molecules have the potential to be targeted for the treatment of chronic pain comorbid with anxiety and depression

### 2.2. GABA Neurons

A genetic study of pain intensity showed that gene expression has significant tissue-group enrichment, such as histone modification in the human midbrain, specifically in GABAergic neurons [39]. GABA neurons in the midbrain serve as a modulator of pain [40]. Functionally, GABA neurons have been reported to be antianalgesic in the DRN. Some studies have indicated that GABA neurons in the DRN are activated by noxious stimuli, such as plantar electrical stimulation and tail pinch [41]. Conversely, other research demonstrated that chemogenetic inhibition of GABA neurons in the DRN promoted nociceptive sensitivity, and activation of GABA neurons alleviated hyperalgesia induced by inflammatory pain or ovarian hormone withdrawal [40,42]. However, a contradiction is presented regarding the involvement of GABA neurons in pain modulation within the DRN. Given that synapses between GABA and 5-HT neurons are inhibitory, it has been hypothesized that the activation of GABA neurons may actually promote pain sensitization, revealing a potential contradiction in their role. Additionally, in vivo electrophysiological studies have shown that morphine directly activates 5-HT neurons without altering GABA neuron activity [43]. However, local GABAergic afferents can exert strong inhibitory postsynaptic currents on 5-HT neurons, potentially suppressing their activity [44,45]. These finding suggest that different neuronal microcircuits may be active in different states. This behavioral outcome may be attributed to the dynamic network activity during progressive pain. Because it is closely associated with some pain-related brain regions such as CeA, BNST, VTA, and ventrolateral periaqueductal gray (vlPAG) [5,46], future research should place greater focus on the role of GABA neurons within the DRN.

### 2.3. DA Neurons

DA neurons are the smaller population in the DRN, which do not overlap with 5-HT and GABA neurons and project to several brain regions, including the BNST, CeA, and VTA, through D1 and D2 dopamine receptors [47,48]. In addition to memory expression, drug addiction, and sleep rhythm, DA neurons are also involved in pain modulation [49,50]. DA neurons in DRN contribute to sex differences in pain-related behaviors [51]. Activation of DA neurons within DRN or vlPAG/DRN DA^+^ terminals in the BNST reduces nociceptive sensitivity induced by inflammatory pain in male mice, whereas activation of this pathway in female mice leads to increased locomotion in the presence of salient stimuli [52]. DA neurons corelease glutamate, and their role is similar to glutamatergic neurons in antinociception [53]. In addition, DA neurons within DRN play a role in opioid-related processes. They are disinhibited by opioid receptor agonists to attenuate nociceptive sensitivity [49,54,55] and activated by ethanol to reduce nociceptive sensitivity [52]. Opioids can directly activate mu-opioid receptors (MOR) or inhibit GABA release to disinhibit the activity of DA neurons. This modulation ultimately increases the release of dopamine and glutamate, which act on downstream targets such as the BNST to mediate analgesic effects [56,57]. Results have shown that the activity of DA neurons in DRN changes in response to pain, and these changes can mediate pain-related hyperalgesia by affecting dopamine synthesis and downstream neuronal circuits.

### 2.4. Glutamatergic Neurons

Cells within the DRN are glutamatergic, expressing the mRNA of either vesicular glutamate transporter 2 (VGLUT2) or VGLUT3. Among these, VGLUT3 is primarily found in 5-HT neurons within DRN. It has been observed that the basolateral amygdala (BA) receives dense innervation from DRN 5-HT neurons and is VGLUT3-positive. The DRN^5-HT∩vGluT3^-BA pathway is associated with social stimuli and anxiety-related stimuli [58]. VTA, as an award center, receives dense glutamatergic and serotonergic input from the DRN [59]. However, chronic pain increased the activity of DRN VGLUT3-positive glutamatergic, but not 5-HT neurons, projecting to the VTA in male mice, and inhibition of the DRN^Glu^-VTA pathway produced an analgesic effect in male mice [60,61]. In addition, there are SP receptors on glutamatergic neurons [62]. Substance P released from central terminals in the brain may act on these receptors, leading to the direct or indirect modulation of glutamate release. This interaction between SP and glutamate systems potentially plays a role in the regulation of pain.

### 2.5. Other Types of Neurons

In addition to glutamate and substance P, acetylcholine and mygalin have been reported to be involved in pain modulation. Specifically, microinjection of the cholinergic agonist carbachol into the DRN has been shown to produce a strong analgesic effect [63]. Additionally, a low concentration of mygalin into DRN can increase the threshold, while a high concentration of mygalin could decrease it. The mechanism behind this modulation may involve the polyamines acting on excitatory and inhibitory neurotransmission simultaneously [64].

## 3. Afferent and Efferent Connections

Since 5-HT and GABA neurons constitute approximately 90% of the total neuronal population in the DRN, our focus has been on the afferent and efferent connections of these two neuron types. The findings are as follows.

### 3.1. Afferent Connections

Whole-brain screening studies in mice identified that GABA neurons in DRN receive fiber input mainly from the hypothalamus, midbrain, and pons, including the somatic motor area (MO), NAc, BNST, lateral stiff nucleus (LH), CeA, vlPAG, midbrain reticular and nucleus (MRN), VTA, SNc, and pons-motor-related area (P-mot) [5,65,66]. Interestingly, several researchers have found that 5-HT neurons mainly received projections from PAG, MRN, LH, P-mot, substantia nigra pars reticulata (SNr), zona incerta (ZI), and IC [7,67], which are similar to the afferent brain regions of GABA neurons in the DRN, indicating that the two types of neurons may encode and regulate the same function. However, there are some differences between these two types of neurons. 5-HT neurons in the DRN receive preferential projections from cerebral cortex regions such as the medial prefrontal cortex (mPFC) and ACC [27,68]. On the other hand, GABA neurons in the DRN receive preferential projections from the CeA and VTA, reflecting the heterogeneity of neurons [59]. These differences indicate the heterogeneity of neurons within the DRN and suggest that the 5-HT and GABAergic systems may play distinct roles in pain processing and related functions.

### 3.2. Efferent Connections

The upstream fiber of the DRN is the main part of the efferent fiber. They are divided into three groups: dorsal group, middle group, and ventral group [69]. The dorsal group of efferent fibers from the DRN travels through the PAG and paraventricular thalamic nucleus (PVT). The medial bundle of DRN in the middle group is radially distributed through the ventrolateral side, and the ventral group travels in the VTA and the medial forebrain. 5-HT neurons can project upward to VTA and SNc in the midbrain. They also project along the medial forebrain tract (MFB) to many forebrain regions such as hypothalamus, thalamus, CeA, striatum (Str), Nac, and cortex [1,70,71], and can also project downward to the posterior brain region. The upstream efferent fibers of GABA neurons are more concentrated. Most of them establish local connections such as PAG and MRN. Some have long-range projections reaching distant brain structures, such as CeA and PVT, which, in turn, innervate back the DRN neurons. Studies have demonstrated that DRN 5-HT neurons can project to VTA, ACC, IC, and CeA to modulate pain [30,72,73]. They also can inhibit the activity of the spinal dorsal horn through synaptic relations with the nucleus raphe magnus (NRM) and also reach the spinal dorsal horn directly to inhibit the transmission of pain information [74]. DRNs are also related to opioids. Green light exposure would activate the projections from the ventral lateral geniculate nucleus (vLGN) proenkephalin (Penk)-positive neurons to the DRN by u-receptor producing analgesic effects [75].

## 4. Interaction between GABA Neurons and 5-HT Neurons

The activity of 5-HT neurons is adjusted by three main inhibitory mechanisms. The first one is that GABA neurons act on 5-HT neurons through GABA_A_ receptors (GABA_A_Rs) and GABA_B_ receptors (GABA_B_Rs) to suppress the activity of 5-HT neurons. The GABA_A_Rs are mainly distributed in the dorsal and dorsolateral fibers and terminus, while GABA_B_Rs are mainly located in the neuronal cell bodies and dendrites [76]. Optogenetic activation of GABA neurons in DRN and 5-HT neurons produced inhibitory postsynaptic currents (IPSCs), which were completely abolished by the tetrodotoxin (TTX) or GABA_A_Rs antagonists (gabazine), but then reappeared when 4-aminopyridine (4-AP) was used in the presence of TTX, confirming that the direct synaptic connections between 5-HT and GABA neurons and GABA acted on 5-HT neurons through GABA_A_Rs [77]. Interestingly, knockout of alpha 1-GABA_A_Rs of DRN, as the most highly expressed subunit in 5-HT DRN neurons, increased the sensitivity of anxiety-like behaviors [78]. In addition, recent researchers have detected that GABA_B_Rs were found on the synaptic membrane containing 5-HT granules under immunoelectron microscopy, and after microinjection of baclofen into the DRN, the 5-HT release decreased significantly [79,80]. This suggests that GABA can regulate 5-HT neurons through GABA_B_Rs. The second is a negative feedback mechanism. The axon branches or dendrites of 5-HT neurons release serotonin, which then activates potassium channels via the 5-HT1A autoreceptors to reduce the excitability of 5-HT neurons [81,82]. 5-HT KO mice also showed increased responsiveness of DRN 5-HT1A autoreceptors [83]. The third is regulated by some neuropeptides such as corticotropin-releasing factor (CRF). When animals cope with environmental challenges like pain, stress, or drug withdrawal, CRF levels increase in the brain, indirectly inhibiting 5-HT neurons by acting on CRF-1 receptors located on GABA neurons [84,85]. The activation mechanisms are regulated by glutamate and norepinephrine (NE). In most animals, 5-HT neurons spontaneously discharge in vivo, but they are silent in vitro. However, they can be evoked in vitro by glutamate and NE released from the locus coeruleus (LC). This activation occurs by inhibiting certain types of potassium (K^+^) channels, such as KCNQ/M and SK-type channels [86].

5-HT neurons also modulate the activity of GABA neurons [70]. GABA neurons express various serotonin receptors, including 5-HT1ARs, 5-HT2Rs, and 5-HT7Rs. Electrophysiological studies have revealed that the activation of 5-HT1ARs reduces the firing frequency of GABA neurons, whereas the activation of 5-HT2Rs and 5-HT7Rs increases the firing frequency of GABA neurons [66,87,88].

Functionally, studies have shown that GABA neurons and 5-HT neurons have opposing behavioral responses. For example, the calcium signal of GABA neurons decreased and the activity of 5-HT neurons increased in rewarding behavior [89]. Conversely, the activity of GABA neurons increased sharply when exposed to aversive stimulation such as plantar shock, whereas the activity of 5-HT neurons showed an inhibited tendency [40]. PFC to DRN would reach GABA neurons preferentially, and produce a rapid inhibition of 5-HT neurons to modulate depression-related behaviors [90]. These also prove that GABA regulates 5-HT expression.

5-HT neurons in DRN also have fiber connections with DA neurons. Research has found that dopamine could cause the depolarization of 5-HT neurons in DRN, and this effect could be antagonized by the D2 receptor antagonist sulpride [91]. Guiard [92] used 6-hydroxydopamine (6-OHDA) to selectively injure DA neurons, and observed that the spontaneous discharge rate of DRN 5-HT neurons in rats was significantly reduced. Therefore, we conclude that the excitation of DA to DRN 5-HT neurons is due to the facilitation of D2 receptors.

In summary, the excitability of DRN neurons is affected by local and external factors, but determining what kind of pathophysiological mechanism it uses in pain regulation needs further study.

## 5. Conclusions

In summary, the past decade has seen exponentially growing interest in the DRN in exploring the neural circuits and regulation of pain information. The DRN is a key hub for transmitting nociceptive signals from the spinal cord, ascending through the specific pathways to the cerebral cortex via the thalamus, and it plays a crucial role in inhibiting pain transmission. The detailed investigations into the molecular biological mechanism of GABA neurons’ interaction with 5-HT neurons have provided new insights into the dysfunction of the DRN. Despite these advancements, many important questions still remain unanswered. For instance, what is the relationship between the DRN and other pain-regulating brain regions such as the vlPAG, VTA, and thalamus? How does the excitability of 5-HT neurons change during the transition from acute to chronic pain? Future research that combines the identification of neuronal subtypes with project-specific targeting and in vivo single-cell recording hold promise for shedding new light on these unresolved questions, potentially leading to a deeper understanding of the DRN’s complex role in pain regulation.

## Figures and Tables

**Figure 1 brainsci-14-00982-f001:**
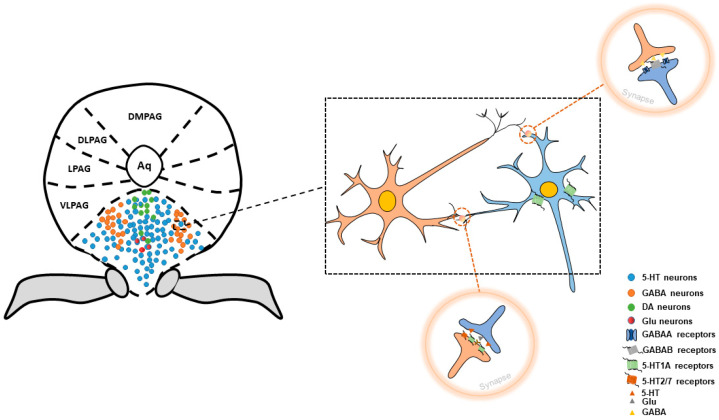
Neurons of distribution in DRN. Neurons in the DRN mainly use serotonin as neurotransmitters, which are predominantly in the midline region. GABAergic neurons are second, being scattered on both sides of the wing. Dopaminergic neurons are close to the midbrain periaqueductal gray. Most glutamatergic neurons are colabeled with 5-HT neurons. There are interactions between GABAergic neurons and serotonergic neurons.

**Figure 2 brainsci-14-00982-f002:**
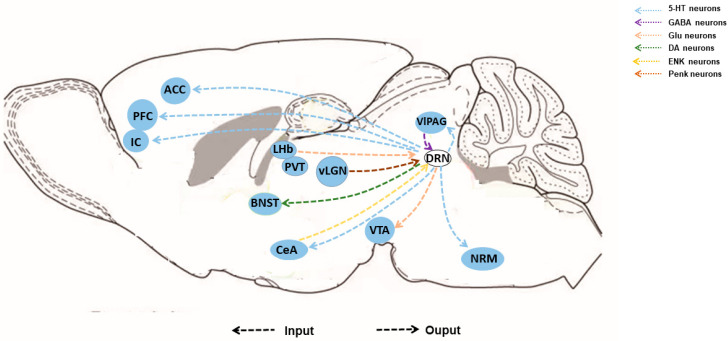
Afferent and efferent connectivity networks of DRN involved in pain regulation. On the output side, serotonergic neurons in the dorsal raphe nucleus (DRN) projecting to the prefrontal cortex (PFC), anterior cingulate cortex (ACC), central amygdala (CeA), and insular cortex (IC) modulate chronic pain comorbid anxiety and depression. Dopaminergic neurons in the DRN projecting to the bed nucleus of the stria terminalis (BNST) mediate analgesic effects. Glutamatergic neurons in the DRN project to the ventral tegmental area (VTA). In addition, DRN sends direct or indirect inputs to the dorsal horn of the spinal cord to inhibit the activity via the raphe magnus nucleus (NRM). On the input side, DRN preferentially receives inputs from GABAergic neurons in the ventrolateral periaqueductal gray (vlPAG), glutamatergic neurons in the lateral habenula (LHb), and proenkephalin-positive neurons in the ventral lateral geniculate nucleus (vLGN) to regulate pain.

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
