# Peer review of "Role of the Dorsal Raphe Nucleus in Pain Processing"

_brainsci, 2024, doi:10.3390/brainsci14100982_

Round 1

Reviewer 1 Report

Comments and Suggestions for Authors

The authors have written a detailed review on neuronal subtypes involved in pain processing in the Dorsal Raphe (DR) based on studies performed on experimental animals.

As a reader I would appreciate of having few examples that may translate to human. In general, SSRIs do not provide clinically meaningful analgesia in clinical trials and therefore they are not recommended for the treatment of human pain. However, number of human genetic studies show that certain psychiatric traits such as neuroticism correlate positively with pain/pain intensity (e.g. PMID: 38429522). Could DR play a role here? High neuroticism appear to correlate with low 5-HT1A receptor binding in all brain regions (PMID: 26337006). Please discuss.

It is unclear what the authors mean here: The levels of 5-HT neurons are adjusted by three main inhibitory mechanisms. The level of electrophysiological firing activity of 5-HT neurons...?Comments on the Quality of English Language

English is readable. Final language check prior publication recommended.

Author Response

  1. As a reader I would appreciate of having few examples that may translate to human. In general, SSRIs do not provide clinically meaningful analgesia in clinical trials and therefore they are not recommended for the treatment of human pain. However, number of human genetic studies show that certain psychiatric traits such as neuroticism correlate positively with pain/pain intensity (e.g. PMID: 38429522). Could DR play a role here? High neuroticism appear to correlate with low 5-HT1A receptor binding in all brain regions (PMID: 26337006). Please discuss.

Response: Thank you very much for your advice. According to your suggestion, we have carefully modified the manuscript. SSRIs are the first-line treatment for depression clinically by blocking serotonin reuptake. Although SSRIs have been proposed as an alternative treatment for chronic pain, their safety and efficacy in pain management remain uncertain (added it to the section of ‘2.1. 5-HT neurons’).

As we know, psychiatric symptoms such as neuroticism, anxiety and depression are frequently observed in clinical settings and often create a cycle that exacerbates pain and psychiatric symptoms [PMID:15327811;14609780]. A meta-analysis of genome-wide association for neuroticism found that dopaminergic and serotonergic neurons showed the strongest association with neuroticism (PMID: 29942085). Genetic study of pain intensity has demonstrated that gene expression shows significant tissue-group enrichment, particularly histone modification within the human adult midbrain, specifically involving GABAergic neurons (PMID: 38429522). GABAergic neurons in the midbrain are known to modulate pain [PMID:36526697]. Some studies have indicated that GABAergic neurons in the DRN are activated by noxious stimuli and activation of the GABAergic neurons in the DRN has been shown to effectively alleviate pain [PMID: 36526697; 39106454] (we added it to the section of ‘2.2. GABA neurons’). In summary, the DRN may play a significant role in modulating pain intensity and neuroticism.

In the midbrain raphe nuclei, presynaptic 5-HT1A autoreceptors modulate the firing of serotonergic neurons, and blocking these receptors to dampen this negative feedback system present a potential pharmacological strategy for treating major depression [PMID:36302033]. The 5-HT1A receptor is implicated in various mood and anxiety disorders, with imaging studies showing reduced binding in cases of major depression [PMID:26851834]. Hirvonen study found an inverse correlation between the personality trait neuroticism and the 5-HT1A receptor binding potential (BPp) in healthy subjects who scored high on the neuroticism factor had lower 5-HT1A receptor BPp than those who scored low. [PMID:26337006]. This suggest that a reduction in 5-HT1A receptors density may lead to increase serotonergic neurons firing, which could alleviate neuroticism and depression symptoms.

Additionally, we added few examples that may translate to human. For instance, a study used resting state fMRI to assess the neural circuit mechanisms in patients with chronic pain comorbid with depressive symptoms. The findings showed functional connectivity of the DRN–CM pathway was decreased [PMID: 31451801] (we added it to the section of ‘2.1. 5-HT neurons’).

  1. It is unclear what the authors mean here: The levels of 5-HT neurons are adjusted by three main inhibitory mechanisms. The level of electrophysiological firing activity of 5-HT neurons...?

Response: Thank you very much for your advice. We have modified ‘The levels of 5-HT neurons are adjusted...’ to ‘The activity of 5-HT neurons is adjusted...’. It means that the activity of 5-HT neurons is adjusted by three main inhibitory mechanisms including GABA receptors, 5-HT1A autoreceptors on 5-HT neurons and some neuropeptides.

Reviewer 2 Report

Comments and Suggestions for Authors

1- The abstract contains vague phrasing such as "has been witnessed" when discussing the dorsal raphe nucleus (DRN). The author must state specific discoveries or advancements regarding DRN’s role in pain processing.

2- Ensure consistent formatting of references throughout the manuscript. For instance, some references include page numbers (e.g., "34(6): 575-85"), while others do not.

3- Discussing GABA neurons, a contradiction is presented regarding their role in pain modulation. The paper mentions that "GABA neurons increased significantly in response to pain stimuli," but later states that inhibition of GABA neurons enhances analgesia. This contradiction should be better explained

4- The author needs to expand on the downstream effects of modulating 5-HT and DA pathways, such as specific pain relief mechanisms or clinical implications for chronic pain management

Author Response

  1. The abstract contains vague phrasing such as "has been witnessed" when discussing the dorsal raphe nucleus (DRN). The author must state specific discoveries or advancements regarding DRN’s role in pain processing.

Response: Thank you very much for your advice. We have carefully modified the abstract. As early as 1993, the DRN was identified as the ‘‘pain inhibitory nucleus’’ in the brain [PMID: 7922601]. A recent study has found that implanting microelectrodes into the DRN and utilizing deep brain stimulation provides analgesia in patients [PMID: 3462392]. Regarding specific cell subtypes involved in pain modulation, activation of 5-HT and DA neurons in the DRN and the neighboring PAG exerts an antinociceptive effect. Although GABA neurons in the DRN are often considered interneurons that locally regulate the functions of other neurons [PMID: 3017320]. Recent research suggests that GABA neurons in the DRN has been reported to be antianalgesic [PMID: 36526697]. This review summarizes the function of specific cell types within DRN in the pain regulation, aiming to improve understanding of the underlying mechanisms.

2- Ensure consistent formatting of references throughout the manuscript. For instance, some references include page numbers (e.g., "34(6): 575-85"), while others do not.

Response: Thank you very much for your advice. We research the references again and add the page numbers. But we find the page number of some references is "2019, 14(8): e46464" or online ahead of print.

3- Discussing GABA neurons, a contradiction is presented regarding their role in pain modulation. The paper mentions that "GABA neurons increased significantly in response to pain stimuli," but later states that inhibition of GABA neurons enhances analgesia. This contradiction should be better explained

Response: Thank you very much for your advice. Functionally, GABA neurons in the DRN have been reported to exert antianalgesic effect. Fiber-photometry imaging showed that GABA neurons activity in the DRN significantly increased in response to pain stimuli and activation of these neurons alleviates hyperalgesia induced by inflammatory pain or ovarian hormone withdrawal (PMID: 36526697; 39106454). However, contradictory finding exist regarding the role of GABA neurons in pain modulation within the DRN. The synapses between GABA neurons and 5-HT neurons have inhibitory effect, activation of GABA neurons might actually sensitize pain. Additionally, a vivo electrophysiological studies have shown that morphine directly activates 5-HT neurons without altering GABA neuron activity (PMID: 34762982). Nonetheless, local GABAergic afferents can exert strong inhibitory postsynaptic currents on 5-HT neurons, potentially suppressing their activity (PMID: 28329692; 36161495). These findings suggest that distinct neuronal neuronal microcircuits may be engaged under different conditions, and the behavioral outcome might reflect dynamic network activity during progressive pain.

4- The author needs to expand on the downstream effects of modulating 5-HT and DA pathways, such as specific pain relief mechanisms or clinical implications for chronic pain management

Response: Thank you very much for your advice. We have added on the downstream effects of 5-HT and DA pathways in modulating pain. DA neurons, which form a smaller population in the DRN, project to various brain regions, including the BNST, CeA, and VTA, acting through D1 and D2 dopamine receptors. The DA neurons contribute to sex differences in pain-related behaviors. Activation of DA neurons within the DRN or vlPAG/DRN DA+ terminals in the BNST reduces nociceptive sensitivity induced by inflammatory pain in male mice. However, in female mice, activation of this pathway leads to increased locomotion in the presence of salient stimuli. DA neurons co-release glutamate, functioning similarly to glutamatergic neurons in anti-nociception. Furthermore, DA neurons in the DRN are involved in opioid-related processes. They are disinhibited by opioid receptor agonists to attenuate nociceptive sensitivity and activated by ethanol to reduce nociceptive sensitivity (we added it to the section of ‘2.3. DA neurons’).

5-HT neurons within DRN are the primary source of forebrain 5-HT. In the central regulation of descending pain facilitation/inhibition from the DRN, various brain regions, such as cortex, thalamus and midbrain, are involved in the DRN pain processing function. In sciatic nerve ligation model, the activity of DRN 5-HT neurons increased, leading to elevated levels of 5-HT release in the PFC. Optical stimulation of DRN 5-HT–PFC pathway alleviated sleep dysregulation induced by chronic pain (PMID: 2437023). Some research found a neural circuit between DRN and ACC modulated chronic pain-induced dysfunction. Activation of the ACC-DRN5-HT pathway reversed chronic pain-induced anxiety-like behaviors (PMID: 38018559) and activation of DRN5-HT– ACC pathway reversed sociability deficits (PMID: 34080539). Studies have measured serotonin release in the CeA and ACC. The neural activity of 5-HT neurons increased after formalin injection and increased 5-HT concentrations in the CeA and ACC [PMID: 37047627]. In mice with spared nerve injury, the firing of 5-HT neurons in the DRN decreased and resulted in disinhibition of CeASOM neurons. Activation of the DRN5-HT–CeASOM pathway relieved pain and depression-like behaviors. The study also used resting state fMRI to assess the neural circuit mechanisms for patients with chronic pain comorbid with depressive symptoms. The results showed functional connectivity of the DRN–CM pathway was decreased [PMID: 31451801]. These findings showed that 5-HT neurons in the DRN encode chronic pain by distinct pathway (we added it to the section of ‘2.1. 5-HT neurons’).